# Taste Changes in Patients with Inflammatory Bowel Disease: Associations with PROP Phenotypes and polymorphisms in the salivary protein, Gustin and CD36 Receptor Genes

**DOI:** 10.3390/nu12020409

**Published:** 2020-02-04

**Authors:** Melania Melis, Mariano Mastinu, Giorgia Sollai, Danilo Paduano, Fabio Chicco, Salvatore Magrì, Paolo Usai, Roberto Crnjar, Beverly J. Tepper, Iole Tomassini Barbarossa

**Affiliations:** 1Department of Biomedical Sciences, University of Cagliari, 09042 Monserrato (CA), Italy; melaniamelis@unica.it (M.M.); gsollai@unica.it (G.S.); crnjar@unica.it (R.C.); 2Department of Medical Sciences and Public Health, University of Cagliari, Presidio Policlinico of Monserrato, 09042 Monserrato (CA), Italy; danilo.paduano@libero.it (D.P.); Chicco.fabio90@gmail.com (F.C.); salvo10ms@libero.it (S.M.); paolo.usai@unica.it (P.U.); 3Department of Food Science, School of Environmental and Biological Sciences, Rutgers University, New Brunswick, NJ 08901-8520, USA; btepper@sebs.rutgers.edu

**Keywords:** inflammatory bowel disease (IBD), taste, gene effects

## Abstract

Inflammatory bowel disease (IBD) is a chronic inflammatory condition of the gastrointestinal tract resulting from interactions among various factors with diet being one of the most significant. IBD-related dietary behaviors are not clearly related to taste dysfunctions. We analyzed body mass index (BMI) and perception of six taste qualities and assessed effects of specific taste genes in IBD patients and healthy subjects (HC). BMI in IBD patients was higher than in HC subjects. Taste sensitivity to taste qualities was reduced in IBD patients, except for sour taste, which was higher than in HC subjects. Genetic variations were related to some taste responses in HC subjects, but not in IBD patients. Frequencies of genotype AA and allele A in *CD36* polymorphism (*rs1761667*) were significantly higher in IBD patients than in HC subjects. The taste changes observed could be explained by the oral pathologies and microbiome variations known for IBD patients and can justify their typical dietary behaviors. The lack of genetic effects on taste in IBD patients indicates that IBD might compromise taste so severely that gene effects cannot be observed. However, the high frequency of the non-tasting form of *CD36* substantiates the fact that IBD-associated fat taste impairment may represent a risk factor for IBD.

## 1. Introduction

Inflammatory bowel disease (IBD) is a chronic and relapsing inflammatory pathology of the gastrointestinal tract. The main clinical phenotypes are Crohn’s disease (CD) and Ulcerative Colitis (UC). Both disorders are characterized by flare-ups, followed by periods of remission. In the past several decades, the incidence and prevalence of UC and CD have increased worldwide, especially in western countries, in concert with industrialization and urban lifestyle [1,2]. The inflammatory processes associated with IBD normally lead to significant weight loss [3], however some IBD patients gain excessive weight masking nutrition issues, such as nutritional deficiencies [4]. IBD is the result of a complex interaction between genetic, immunologic, microbial, and environmental factors [5,6,7,8]. However, its precise aetiology remains uncertain [6].

Accumulating evidence suggests that diet is one of the most significant risk factors that contribute to the pathogenesis of IBD [6]. Some studies found an increased risk of IBD associated with the consumption of foods that are high in sugar, animal fats and oils, and protein [1,6,9,10,11,12,13,14]. On the other hand, high intake of foods rich in fiber, including fruits and vegetables, has been shown to protect against IBD [9,10,11,15,16,17]. In addition, it is known that some nutrients (such as Tryptophan, Arginine, Glutamine and short-chain fatty acids derived from dietary fiber) enhance intestinal barrier function and host immunity, which in turn protects against this disease [6]. Accordingly, diet recommendations that encourage the consumption of plant-based foods and de-emphasize excess consumption of sugar and animal foods may be a successful strategy for treatment of IBD.

Systemic inflammation and compromised immunity are characteristic features of IBD that put these patients at greater risk for oral diseases such as dental caries and periodontitis [18]. Thus, it is plausible that IBD affects overall taste function as well. In fact, Schutz et al. have suggested that IBD patients overconsume sweets, fats, and proteins as a way to compensate for taste impairments and thus to make their meals more satisfying [19]. However, published data on taste changes in IBD patients are controversial [4,19,20,21,22,23]. Some authors have reported a reduction in overall taste function [4] or reductions limited to specific qualities [4,19,20], while others reported no taste impairments in these patients [23]. In addition, some studies reported that IBD patients with alterations of taste function show zinc deficiency [19,22,24,25], which may be linked to the functionality of the salivary zinc-dependent enzyme gustin/carbonic anhydrase VI (CAVI) [19]. Gustin CAVI is considered a trophic factor promoting development of taste buds [26,27,28] and disruptions in this protein are known to decrease taste function [29]. Importantly, the extant literature on taste changes in IBD has focused on the classic four taste qualities (sweet, salty, bitter, sour), but studies have not included the fifth basic taste (umami) or the ability to taste fatty acids which has been recently proposed as a sixth sensory quality [30]. Evidence supporting fat taste as an oral sensation is based on evidence showing that the primary transduction mechanism for fat taste consists of the interaction of saturated and unsaturated long-chain fatty acids (LCFA; number of carbons ≥16, such as linoleic acid) with the plasma membrane glycoprotein CD36 in taste buds [31,32,33,34,35,36]. In fact, the multifactorial CD36 scavenger receptor is an 88kDa protein which has been shown to be present in gustatory papillae of humans [37,38] as the main long-chain fatty acid receptor in taste bud cells where it plays a decisive role in the orosensory perception of dietary lipids (such as oleic acid), and fat preference [39,40,41]. Thus, major gaps still exist in our understanding of taste mechanisms in IBD.

Individual differences in taste responses are well known and are partially controlled by genetic factors. The genetic ability to taste the bitter compound 6-*n*-propylthiouracil (PROP) is the best-known example of taste variability that has broad implications for taste perceptions, food preferences and dietary behavior with subsequent impacts on nutritional status and health outcomes [42]. It is well-established that super-tasters (who perceive extreme bitterness from PROP), compared with PROP non-tasters (who are taste blind to this compound), are more responsive to other taste qualities including fats, and that PROP tasting is associated with variations in food acceptability, selection of vegetables and fruits and several health parameters, such as body mass composition, plasma antioxidant status and colon cancer risk [39,43,44,45,46,47,48,49,50,51,52,53,54,55,56]. The observation that PROP super-tasters are more responsive to a wide range of oral stimuli, compared to PROP non-tasters, is explained by the greater density of fungiform taste papillae on the anterior surface of the tongue that has been reported in these individuals [57,58,59,60,61]. In addition, PROP sensitivity has been associated with the polymorphism, *rs2274333* (A/G), of the gustin gene (CA6) which alters the functionality of gustin CAVI as trophic factor of taste papillae by modifying its zinc binding capacity [27,28].

It is not known if taste variations associated with the aforementioned genes, as well as polymorphisms in *CD36* (the gene implicated in fatty acid taste), play a role in taste changes in IBD. Investigation of these gene effects could provide important insights for linking taste changes to dietary behavior in IBD.

The aim of this study was evaluating whether differences in body mass index (BMI), in IBD patients, may be associated with specific alterations in taste function. Given that, in IBD patients, manifold oral pathologies have been observed to be caused by iron, zinc, or vitamin deficiencies [62,63,64], we hypothesized specific taste dysfunctions in IBD patients compared with healthy controls. We further hypothesized a higher BMI in IBD patients compared with controls, based on previous observations showing that patients with IBD overconsume foods rich in sugar, fat and protein [19,20,21]. To achieve this aim we determined if IBD patients differed from healthy controls in BMI and perception of all six taste qualities (sweet, salty, sour, bitter, umami and fat). We also assessed gene effects on these responses with the aim to understand the mechanisms involved in the possible alterations of taste of IBD patients. Subjects were classified for their PROP taster status and genotyped for (a) the *rs2274333* (*A/G*) polymorphism of the gustin gene; and (b) the *r1761667* polymorphism of the *CD36* gene. Previous studies demonstrated a role for this *CD36* gene variation in the protein expression levels, in fat perception and metabolism [39,41,65,66,67,68,69].

## 2. Materials and Methods

### 2.1. Subjects

One hundred and fifty-nine Caucasian volunteers were recruited in the area of Cagliari, Italy. Two groups were studied: inflammatory bowel disease (IBD) patients (*n* = 97; 53 men, 44 women; age 51.38 ± 1.5 y) and healthy control (HC) subjects (*n* = 62; 26 men, 36 women; age 48.79 ± 3.06 y). IBD patients were referred to the study by the IBD clinics of the Gastroenterology Unit of the University Hospital Company (AOU) Monserrato (CA), Italy and included Crohn’s Disease (CD) (*n* = 43) and Ulcerative Colitis (UC) patients (*n* = 54). HC were recruited at the local University and matched in age to the clinic population. All patients enrolled (both CD and UC patients), had a disease in remission and were treated with mesalazine or 5-ASA agents or monoclonal antibodies against TNF-α for their disease. Table 1 shows the demographic and clinical characteristics of IBD patients. BMI in IBD patients was stable and did not tend to change over time, because of the condition of disease remission. Exclusion criteria for both IBD patients and HC subjects were otolaryngology disorders, major systemic diseases, drug interfering with taste or smell (e.g., steroids, antihistamines, and certain antidepressants), pregnancy or lactation, food allergies, history of middle ear surgery, cranial trauma, Bell’s palsy, or stroke. IBD patients who had any systemic diseases associated with IBD were not included. Procedures were carried out in accordance with the Helsinki Declaration and approved by the ethical committee of AOU of Cagliari. Subjects signed an informed consent prior to being enrolled in the study.

### 2.2. Experimental Procedure

Subjects were requested to refrain from consuming any food or beverages, and from smoking, using chewing gum or oral care products for at least 2 h prior to taste tests. They had to be in the test room 15 min before the beginning of the session in order to adapt to the environmental conditions (23–24 °C; 40–50% relative humidity). Weight and height were measured for each subject in order to calculate the BMI (Kg/m^2^) and a sample of whole saliva (2 mL) was collected into an Eppendorf tube. The samples were stored at −80 °C until the molecular analyses were completed as described below.

For all taste assessments (see the taste assessment section), the solutions, prepared the day before the session, were stored in a refrigerator until 1 h before testing. Stimuli were administered at room temperature.

### 2.3. PROP Taster Status Classification

Subjects were classified for PROP taster status by using the impregnated paper screening test [70], which was validated in numerous studies [71,72]. Briefly, two paper disks, one impregnated with PROP solution (50 mmol/L) and one with sodium chloride, NaCl (1.0 mol/L), were placed on the tip of the tongue of each subject for 30 s. The evaluations of the perceived intensity for each paper disk were determined by using the LMS scale [73]. The use of this scale gave the subjects the freedom to estimate the PROP bitterness intensity relative to the “strongest imaginable” oral stimulus ever perceived in their life. The LMS is a semi-logarithmic 100-mm scale in which seven verbal descriptors are arranged along the length of the scale. Subjects who gave to the PROP disk lower ratings than 15 mm on the LMS were classified as PROP non-tasters (NT), those who gave higher ratings than 67 mm on the LMS were classified as PROP super-tasters (ST), while medium-tasters rated PROP disk with intermediate values [70]. For the subjects who gave a borderline rating for PROP disk, the ratings for NaCl were used as a reference standard since the taste intensity to NaCl does not change with PROP taster status in this procedure [74]. Based on their taster group assignments 14 HC subjects and 25 IBD patients were classified as ST, 29 HC subjects and 39 IBD patients were MT, and 25 HC subjects and 33 IBD patients were NT.

### 2.4. Sweet, Salty, Sour, Bitter and Umami Taste Sensitivity Assessments

Taste sensitivity to the four primary qualities (sweet, sour, salty, bitter) was examined by using the Taste Strip Test (TST, Burghart Company, Wedel, Germany) [75,76]. Briefly, 16 filter paper strips impregnated with four concentrations of each taste quality (0.05, 0.1, 0.2, or 0.4 g/mL of sucrose; 0.05, 0.09, 0.165, or 0.3 g/mL of citric acid; 0.016, 0.04, 0.1, or 0.25 g/mL of NaCl; 0.0004, 0.0009, 0.0024, or 0.006 g/mL of quinine hydrochloride) were presented to each subject sequentially. The subject placed each paper strip on the tongue and identified the taste quality he/she perceived. Each correct answer was rated 1 and the maximum score for the whole TST was 16 (4 per each taste quality). A subject was considered normogeusic if he/she scored ≥9, hypogeusic or ageusic if he/she scored <9 (total taste score below the 10th percentile) [76]. Taste sensitivity for umami was examined using four circular filter papers impregnated with 10 µL of monosodium glutamate (MSG) solutions (0.0017, 0.0085, 0.0170 or 0.0338 g/mL) which were prepared immediately before the test session. Each correct answer was rated 1 and the maximum score was 4. Taste qualities were presented in a pseudo-randomized manner and concentrations within each solution type were tasted from the lowest concentration to the highest. Before each stimulation subjects rinsed the mouth with spring water.

### 2.5. Oleic Acid Threshold Assessment

Oleic acid thresholds were assessed, in the absence of nose clips, by a variation of the staircase method implemented in a 3-Alternative Forced Choice procedure according to Melis et al. [39]. Subjects were presented with three paper filter disks: two impregnated with 10 µL of paraffine (control) and one with the amount of oleic acid under evaluation. Oleic acid samples were presented in ascending order from the lowest concentration (0.0015 µL) to the highest (pure) until subjects were able to identify the odd sample in two consecutive trials. The oleic acid concentration presented was increased after a single incorrect response and reduced after two consecutive correct responses. The concentration at which subjects correctly identified the odd sample was reported as the detection threshold of the subject. All were instructed to rinse their mouths after each triad. The time between triads was approximately 1–2 min. Twenty subjects were not able to distinguish paper disks impregnated with pure oleic acid from those of controls. These subjects were excluded from the oleic acid thresholds and *CD36* molecular analyses.

### 2.6. Molecular Analysis

DNA was extracted from saliva samples by using the QIAamp^®^ DNA Mini Kit (QIAGEN S.r.l., Milan, Italy) according to the instructions. Subjects were genotyped and for the single nucleotide polymorphism (SNP), *rs2274333* (A/G) of gustin gene located in the exon 3 that results in a substitution of amino acid Ser90Gly, and for the *rs1761667* (G/A) SNP of *CD36* gene, located at the—31,118 promoter region of exon 1A. The gustin gene and *CD36* gene regions were amplified by PCR techniques. Amplified samples of the fragments of 253 bp including the polymorphism *rs2274333* (A/G) of gustin gene were digested with restriction enzyme (*HaeIII*) according to Padiglia et al. [28]. The fragments including the *rs1761667* (G/A) SNP of *CD36* were digested by the restriction enzyme (*HhaI*) according to Banerjee et al. 2010 [77]. These methods have been validated by numerous studies [26,27,39,65,67,71,78]. Electrophoresis on a 2% agarose gel was used to separate the products of digestion. DNA bands were visualized by staining with ethidium bromide and ultraviolet light to highlighting the deletion. PCR 50 bp Low Ladder DNA was used as a marker of molecular mass (Gene Ruler™-Thermo Scientific, Waltham, MA, USA).

### 2.7. Data Analyses

One-way ANOVA was used to compare the differences in BMI, total taste score of the whole TST, taste score of each taste quality and oleic acid threshold between IBD patients and HC subjects. Two-way ANOVA was used to analyze the differences in BMI, total taste score of the whole TST, taste score of each taste quality between IBD patients and HC subjects according to PROP taster status and the *rs2274333 (A/G)* polymorphism of the gustin gene and to compare differences in oleic acid threshold between two groups according to PROP taster status and the *rs1761667* polymorphism of *CD36*. Analysis of covariance (ANCOVA) was also used to control for differences in BMI between IBD patients and HC subjects that could influence the taste scores. Specifically, one-way ANCOVA (controlling for BMI) was used to compare differences in total taste score of the whole TST, taste score of each taste quality and oleic acid threshold between two groups; two-way ANCOVA (controlling for BMI) was used to analyze the differences between two groups in total taste score and taste score of each taste quality according to PROP taster status and the *rs2274333 (A/G)* polymorphism of the gustin gene; two-way ANCOVA (controlling for BMI) was also used to analyze the differences in oleic acid threshold between two groups according to PROP taster status and the *rs1761667* polymorphism of *CD36*. ANCOVA confirmed all associations and results are shown in the figures. Post hoc comparisons were performed with the Fisher’s least significant difference (LSD) test, except the assumption of homogeneity of variance was violated, in this case the Duncan test was used. *P* values were adjusted by Bonferroni correction (adjusted *P* = *P* ∙ number of groups being compared). Differences between IBD patients and HC subjects on genotype distribution and allele frequency of the *rs2274333 (A/G)* polymorphism of the gustin gene and the polymorphism of the *CD36* gene were compared by using the Fisher method (Genepop software version 4.2; http://genepop.curtin.edu.au/genepop_op3.html). Fisher’s exact test was used to analyze differences in frequency related to PROP taste status between two groups. Statistical analyses were conducted using STATISTICA for Windows (version 10; StatSoft Inc., Tulsa, OK). The significance level was set at *p* < 0.05.

## 3. Results

### 3.1. BMI Effects

Figure 1 shows mean values (±SE) for BMI determined in IBD patients and HC subjects. The same data are also shown according to PROP taster status and the *rs2274333 (A/G)* polymorphism of the gustin gene. One-way ANOVA showed that BMI of IBD patients was higher than that of HC subjects (*F*_1, 157_ = 27.459, *p* < 0.00001) (Figure 1A). Post hoc comparison showed that BMI of IBD patients was higher than that of HC subjects in each PROP taster group (*p* < 0.045; Duncan’s test adjusted by Bonferroni correction subsequent to two-way ANOVA) (Figure 1B). Post hoc comparison also showed that IBD patients who carried the AA and AG genotype at this gustin gene polymorphism had higher BMI than those of HC subjects with similar genotypes (*p* < 0.0051; Fisher’s test LSD adjusted by Bonferroni correction subsequent to two-way ANOVA) (Figure 1C). In contrast, the BMI of IBD patients who had the GG genotype did not significantly differ from that of HC subjects with GG genotype (*p* > 0.05). No differences in BMI values between CD and UC patients were found (*p* > 0.05) (data not shown).

Also, IBD patients and HC subjects did not differ by PROP taster status classification (χ^2^ = 0.598, *p* = 0.741) (Table 2) or in genotype distribution and allele frequency of the gustin gene polymorphism (genotypes χ^2^ = 1.265, *p* = 0.531; alleles χ^2^ = 1.285, *p* = 0.526 Fisher’s test) (Table 3).

### 3.2. Total Taste Scores

Figure 2 shows mean values (±SE) of the total taste score for the TST determined in IBD patients and HC subjects; the same data are also shown according to PROP taster status and the *rs2274333 (A/G)* polymorphism of the gustin gene. One-way ANCOVA controlling BMI showed that the total taste score of IBD patients was lower than that of HC subjects (*F*_1, 156_ = 4.789; *p* = 0.0301) (Figure 2A). The percentage of subjects who were classified as normogeusic or hypogeusic/ageusic in IBD patients differed with respects that determined in HC subjects (χ^2^ = 6.243, *p* = 0.0125). Specifically, 74% (*n* = 72) of IBD patients were normogeusic, and 26% (*n* = 25) were hypogeusic/ageusic, while 90% (*n* = 56) of HC subjects were normogeusic, and 10% (*n* = 6) were hypogeusic/ageusic.

HC subjects who were classified as ST had higher total taste scores than IBD patients who were ST (*p* = 0.00756; Duncan’s test adjusted by Bonferroni correction subsequent to two-way ANCOVA). Also, HC subjects classified as ST had the highest total taste score than those of NT HC subjects (*p* = 0.026 Duncan’s test adjusted by Bonferroni correction subsequent to two-way ANCOVA) (Figure 2B). No significant differences in total taste score related to PROP taster status were found in IBD patients (*p* > 0.05).

HC subjects who carried the AA and AG genotypes of the gustin gene polymorphism were higher than those of the HC subjects with the GG genotype (*p* < 0.039; Duncan’s test adjusted by Bonferroni correction subsequent to two-way ANCOVA), whereas no significant differences related to the gustin gene polymorphism were found in IBD patients (*p* > 0.05) (Figure 2C). Also, no differences in total taste scores between CD and UC patients were found (*p* > 0.05) (data not shown).

### 3.3. Sweet, Salty, Sour, Bitter and Umami Taste Scores

Figure 3 shows mean values (±SE) of the taste scores for sweet, sour, salty, bitter and umami determined in IBD patients and HC subjects; the same data are also shown according to PROP taster status and the *rs2274333 (A/G)* polymorphism of the gustin gene. One-way ANCOVA revealed that the taste score of IBD patients was lower than that of HC subjects for sweet, salty, bitter and umami (sweet: *F*_1, 156_ = 18.640, *p* = 0.00003; salty: *F*_1, 156_ = 6.1010, *p* = 0.01459; bitter: *F*_1, 156_ = 21.686, *p* = 0.00001; umami *F*_1, 156_ = 10.804, *p* = 0.00126), but was higher in IBD patients for sour (*F*_1, 156_ = 36.663, *p* < 0.00001) (Figure 3A).

Results showed in HC subjects a significant effect of PROP taster status only on bitter scores with ST having higher values than MT and NT (*p* < 0.0207; Duncan’s test adjusted by Bonferroni correction subsequent to two-way ANCOVA), though a similar trend was observed for all qualities. No significant differences related to PROP taster status were found in IBD patients (Figure 3B). Results also showed the following effects of disease on taste score of each taster group (Figure 3B): MT and NT IBD patients showed lower sweet scores than corresponding HC subjects in (*p* < 0.016; Fisher’s LSD test adjusted by Bonferroni correction subsequent to two-way ANCOVA); MT and NT IBD patients showed higher sour scores than corresponding HC subjects (*p* < 0.0288; Duncan’s test adjusted by Bonferroni correction subsequent two-way ANCOVA); ST and NT IBD patients showed lower bitter scores than corresponding HC subjects (*p* < 0.015; Duncan’s test adjusted by Bonferroni correction subsequent to two-way ANOVA).

Two-way ANCOVA on the same data according to the gustin gene polymorphism revealed a significant interaction of the participants’ group (IBD patients/HC subjects) × gustin gene genotype group on the taste score relative to sweet (*F*_2, 152_ = 4.6693; *p* = 0.0179) (Figure 3C). Post hoc comparison showed that HC subjects with A allele (AA and AG) gave higher sweet scores than GG HC subjects (*p* < 0.010; Fisher’s LSD test adjusted by Bonferroni correction subsequent to two-way ANCOVA). Post hoc comparison also showed the following effects of disease on taste score of each gustin gene genotype group (Figure 3C): sweet, bitter and umami scores of IBD patients with AA and AG genotypes were lower than those of the corresponding HC subjects, while scores for sour were higher in the IBD patients than HC subjects, (*p* < 0.026; Fisher’s LSD test adjusted by Bonferroni correction subsequent to two-way ANCOVA). No differences between CD and UC patients were found (*p* > 0.05).

### 3.4. Oleic Acid Threshold and Genotyping for CD36 Polymorphism, rs1761667 (A/G).

Figure 4 shows mean values (±SE) of the oleic acid threshold determined in IBD patients and HC subjects; data are also shown according to PROP taster status and the *rs1761667* (A/G) polymorphism of *CD36*. One-way ANCOVA showed that the oleic acid threshold of IBD patients was higher than that of HC subjects (*F*_1,136_ = 44.779, *p* < 0.000001) (Figure 4A). Post hoc comparison showed that the oleic acid threshold of IBD patients was higher than that of HC subjects in both MT and NT (*p* < 0.00025; Duncan’s test adjusted by Bonferroni correction subsequent to two-way ANCOVA). Also, the oleic acid threshold in IBD patients was higher in each *CD36* genotype group compared to these same genotype groups in HC subjects (*p* < 0.017; Duncan’s test adjusted by Bonferroni correction subsequent to two-way ANCOVA) (Figure 4B,C). No differences of the oleic acid threshold between CD and UC patients were found (*p* > 0.05).

IBD patients and HC subjects differed statistically based on the genotype distribution and allele frequency of the *CD36* polymorphism (Genotypes χ^2^ = 6.001, *p* = 0.049; alleles χ^2^ = 6.099, *p* = 0.047 Fisher’s test) (Table 3). Specifically, the genotype AA and allele A were more frequent in IBD patients, while genotype GG and allele G were more frequent in HC subjects.

## 4. Discussion

Previous studies have reported that patients with IBD overconsume foods rich in sugar, fat and protein [19,20,21]. It is uncertain if these dietary behaviors are related to taste dysfunction that could contribute to higher BMIs observed in some studies in IBD patients [79], but not in others [80,81,82]. Previous investigations have not considered a possible role for genetic taste variation in weight status in IBD. In our study, BMI was higher in IBD patients, as compared to HC subjects. There were no specific effects of PROP taster status or the *rs2274333* (A/G) polymorphism of the gustin gene on BMI. Even if the BMI was marginally higher in subjects who were NT as compared to those who were MT and ST. It is interesting to note that the relationship between the gustin gene polymorphism and BMI appears to be opposite in HC and IBD patients. Melis et al. [27] previously demonstrated higher cell culture production of gustin CAVI from the saliva of healthy normal weight individuals with the AA genotype compared to gustin CAVI from saliva of individuals with the GG genotype. However, Lamy and co-workers [83] showed that the level of gustin CAVI was elevated in morbidly obese women relative to normal weight controls. Based on these observations, one can speculate that AA and AG individuals, who are associated with a higher BMI among IBD patients, produce more gustin CAVI. It is unclear whether this would be related specifically to obesity or to IBD.

Our results also showed that overall taste intensity, as well as intensity to specific qualities such as sweet, salty, bitter and umami were reduced for the IBD patients. In contrast, sour taste was higher in IBD patients than in HC subjects. Together, these results are consistent with data reported by Steinbach et al. [4]. It is worth highlighting that, in the present study, the taste parameters were analyzed controlling for differences in BMI between IBD patients and HC subjects. Therefore, the taste changes we observed in IBD patients presumably were attributable to the disease and not to differences in weight between the two groups. The present study also documented, for the first-time, reduced taste perception by IBD patients for umami and fat which are the taste qualities related to appetitive responses to protein- and lipid-rich food sources, respectively.

The overall reduction in taste in IBD patients can be well understood in the light of the manifold oral pathologies observed in these patients [18]. These pathologies can be specific or non-specific manifestations. Moreover, the symptoms that are associated with these oral pathologies (specially acidic taste, taste changes, changes in the tongue and dry mouth) can be caused by iron, zinc, or vitamin deficiencies due to rectal bleeding and intestinal malabsorption linked to IBD, or its pharmacological treatments [62,63,64]. It is also well known that zinc deficiency (regardless of its cause) is associated with taste dysfunction and losses in taste acuity [29].

On the other hand, we observed increased acid taste in IBD patients which could also be related to disturbances in gustin CAVI. This salivary protein is a zinc dependent enzyme that, among other functions, regulates pH balance of the saliva. Low salivary pH promotes the growth of infectious microbes over beneficial ones. The shift in the composition of the oral microbiome observed in IBD patients is associated with increased generation of bacteria-derived acid metabolites and greater risk of oral disease [84]. Therefore, we speculate that disruptions in the levels or in the functionality of gustin CAVI with a consequent reduction of its capacity to neutralize bacteria-produced acids may play a role in the increased sour taste we observed in these patients. These results fit well with data on the role of gustin CAVI in oral health, showing that variations in the gustin gene are associated with the presence of aciduric and acidogenic species which are promoted at low pH [85]. Previous studies in IBD patients indicate low levels of zinc that may be insufficient to activate the gustin enzyme [19,22,24,25]. Our observations further suggest that zinc deficiency in combination with gustin CAVI changes may play a role in the oral dysbiosis in IBD.

Our findings indicate a significant direct relationship between overall taste function or bitter taste and PROP taster status in HC subjects, who displayed a similar trend also for the other qualities. These results are consistent with those revealing that healthy PROP super-taster subjects, compared to non-taster, possess a greater density of fungiform taste papillae that can explain their high general taste sensitivity [57,58,59,60,61]. Differently, we have not found a specific effect of PROP taster status on taste function in IBD patients that may be explained by oral pathologies, especially those in the dorsal surface of tongue [62,63,64]. This suggests that IBD patients who are PROP super-tasters do not have taste advantages over PROP non-tasters.

HC subjects also showed an interesting significant effect of allele A of the gustin locus, which determined a higher overall taste function and a higher sweet taste with respect to allele G. A similar trend was found also for other qualities, except sour. The relationship between gustin CAVI and taste sensitivity is not a matter of consensus, with some data reporting no association between protein levels and taste sensitivity [86]. We studied the role of the *rs2274333* (A/G) polymorphism of the gustin gene on functionality of the protein as a trophic factor of fungiform papillae [26,27,28,71] and found that the substitution of allele A with allele G leads to a structural modification of the protein active site which decreases zinc binding [28]. Therefore, the allele A in the polymorphism results in a functional form of gustin CAVI associated with high cellular proliferation and papilla density, while the allele G leads to a less functional form associated with low cellular proliferation and papilla density. According to these data, the high overall taste function and high sweet taste that we found in HC subjects with at least an allele A support the hypothesis that a high papilla density in these subjects would allow them to perceive taste stimuli more intensely. Rodrigues et al. [87] first studied the relationship between salivary proteins and sweet taste sensitivity. They found an inverse effect showing higher gustin CAVI levels in saliva associated with lower sweet sensitivity. Therefore, further studies are needed to better understand the role of gustin CAVI in sweetness. On the other hand, we did not find a specific effect of gustin gene polymorphism on taste function of IBD patients. These results are consistent with studies reporting IBD patients (with alterations of taste function) to have zinc deficiency [19,22,24,25], and suggest that the low level of zinc in these patients would be insufficient to activate gustin CAVI as a trophic factor for papillae also when it is present in the more functional iso-form.

Contrary to expectations, we observed no specific effects of the genetic factors on taste function of IBD patients. One potential explanation for this finding could be that the damage to the taste system is so severe in IBD, that the effects of genes cannot be observed. This is not the case in obesity (uncomplicated by other diseases) [49,50], in upper respiratory diseases [88,89,90] and different types of cancer [54] where taste genes seem to play a prominent role in disease risk and disease symptoms. For example, *TAS2R38* polymorphisms are strongly related to upper respiratory infections and the risk of chronic rhinosinusitis [88,89,91]. In this respect, IBD seems to be fundamentally different from these other diseases.

Particular attention should be given to the marked decrease of fat perception that we found in IBD patients with respect to HC subjects. This reduction does not depend on the PROP taster status or the *r1761667* polymorphism of *CD36* gene, whose allele G has been already associated with a high expression of CD36 protein [68,69] or an increased taste perception to fat [39,40,41,92,93]. Differently, in HC subjects we previously showed a direct relationship between fatty acid perception and PROP taster status or the polymorphism of *CD36* locus [39]. These data seem inconsistent with respect to what we found in this study. However, the wide differences we observed between HC subjects and IBD patients dampened the effects of the two genetic factors in HC subjects, when the two groups were analyzed together. In fact, if the data of the two groups were analyzed separately, it would be possible to highlight the effect on fat perception of PROP taster status (with PROP super-tasters displaying higher values than non-tasters; *p* = 0.045; Duncan’s test subsequent to one-way ANCOVA), or *CD36* gene (with GG subjects showing higher values than AA subjects; *p* = 0.033; Duncan’s test subsequent to one-way ANCOVA) in HC subjects, but not in IBD patients (*p* > 0.05; Duncan’s test subsequent to one-way ANCOVA).

Our results also show a higher frequency of IBD patients with genotype AA and allele A in *CD36* locus, as compared to HC subjects. This could explain the low sensitivity for fat that we found in these patients since the genotype AA and allele A are associated with low expression of the protein [68,69] or decreased perception to fat [39,40,41,92,93].

## 5. Conclusions

In conclusion, our findings show impaired sweet, salty, bitter, umami and fat taste in IBD patients, but increased sour taste. Reductions in sweet, umami, and fat perception may contribute to increased consumption of foods rich in sugar, fat and protein by IBD patients and possibly excess weight gain in some of these individuals [1,6,9,10,11,12,13,14]. Elevated sour taste in IBD patients may interfere with food enjoyment and may represent a separate quality of life issue in this disease.

In addition, the high frequency of non-tasting form of CD36, which has not been previously described associated with IBD, may substantiate the fact that disruption of fat perception in IBD patients may represent a risk factor for this disease.

## 6. Limitations and Future Directions

This work provides important insights for linking taste changes to dietary behavior in IBD. However, more studies are needed to assess directly relationships between oral sensations and food selection in IBD.

In addition, our findings warrant a closer look at the potential role of this gustin gene polymorphisms in body weight status in IBD, especially in a larger study sample. Our observations further suggest that zinc deficiency in combination with gustin CAVI changes may play a role in the oral dysbiosis in IBD. This interpretation is speculative and should be tested in future studies.

## Figures and Tables

**Figure 1 nutrients-12-00409-f001:**
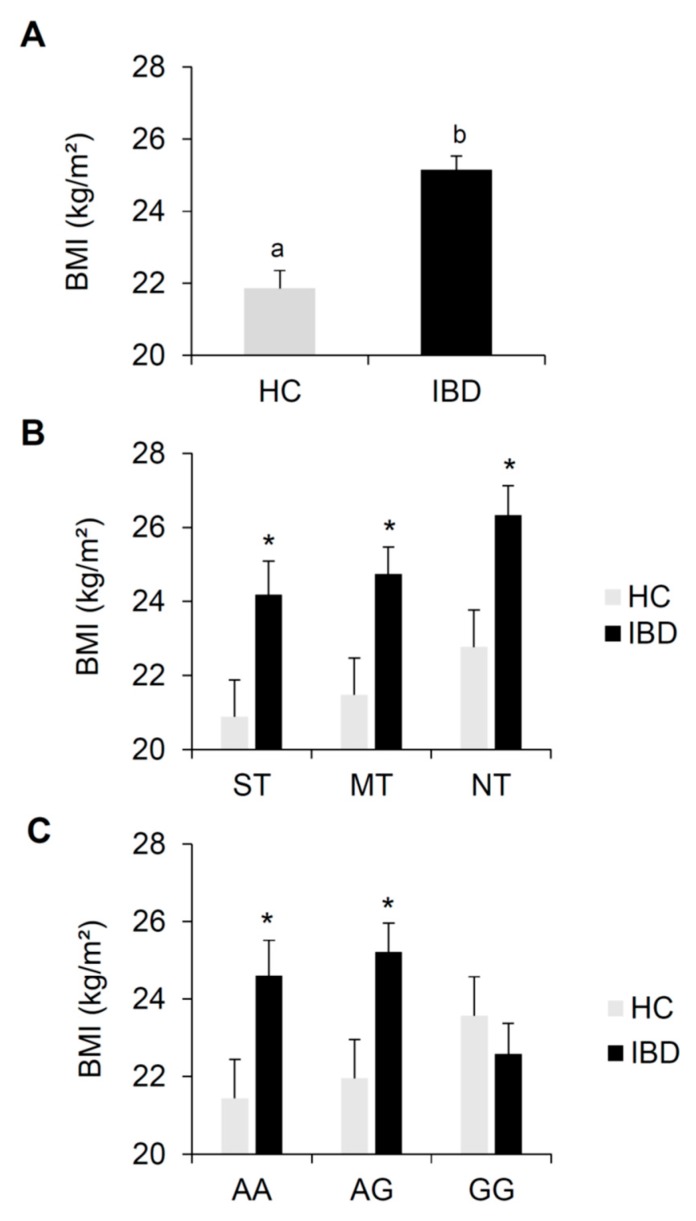
Mean (± SE) values of body mass index BMI (kg/m^2^) determined in HC subjects (*n* = 62) and IBD patients (*n* = 97) (**A**). The same data are shown for each PROP taster group (**B**) and for each genotype group of the *rs2274333 (A/G)* polymorphism of gustin gene (**C**). Different letters indicate a significant difference (A: *F*_1, 157_ = 27.459, *p* < 0.00001, one-way ANOVA). * indicate a significant difference with respect to the corresponding value of HC subjects (B: *p* ≤ 0.045, Duncan’s test adjusted by Bonferroni correction subsequent two-way ANOVA; C: *p* ≤ 0.0051, Fisher LDS test adjusted by Bonferroni correction subsequent two-way ANOVA).

**Figure 2 nutrients-12-00409-f002:**
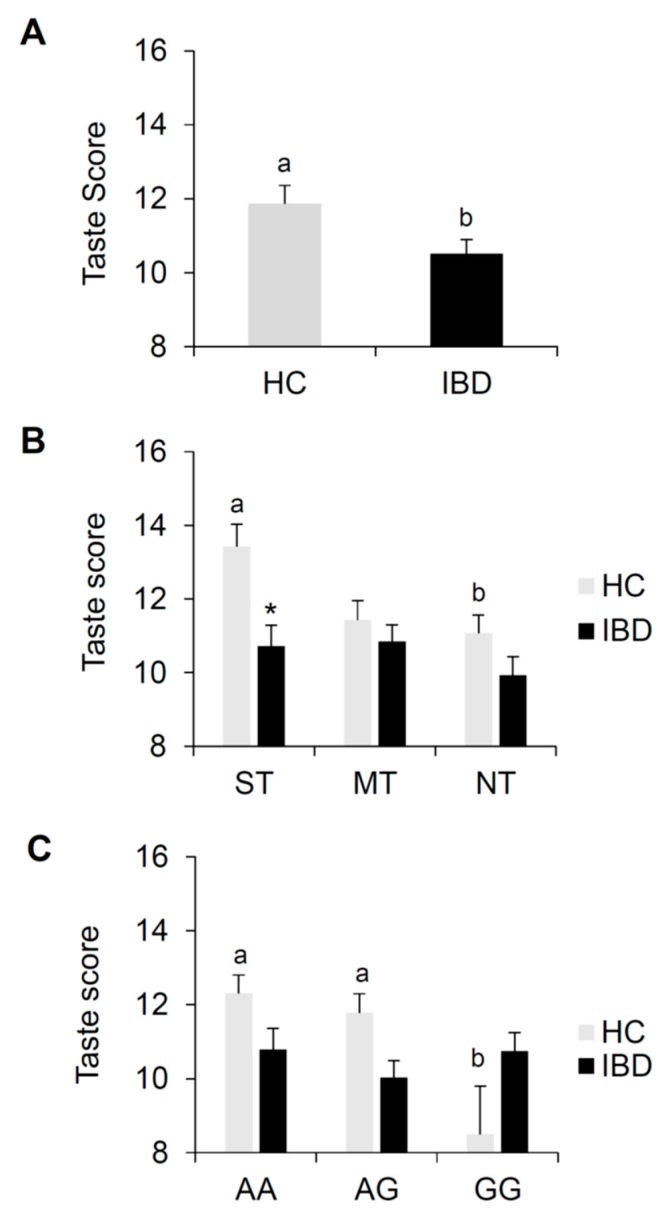
Means (±SE) values of the total taste score of the whole TST determined in IBD patients (*n* = 97) and HC subjects (*n* = 62) (**A**). The same data are shown for each PROP taster group (**B**) and for each genotype group of the *rs2274333* (A/G) polymorphism of gustin gene (**C**). Different letters indicate a significant difference (A: *F*_1, 156_ = 4.789; *p* = 0.0301, one-way ANCOVA; B: *p* = 0.026, Duncan test adjusted by Bonferroni correction subsequent two-way ANCOVA; C: *p* ≤ 0.039, Duncan test adjusted by Bonferroni correction subsequent two-way ANCOVA). * indicate a significant difference with respect to the corresponding value of HC subjects (*p* = 0.0075; Duncan test adjusted by Bonferroni correction subsequent two-way ANCOVA).

**Figure 3 nutrients-12-00409-f003:**
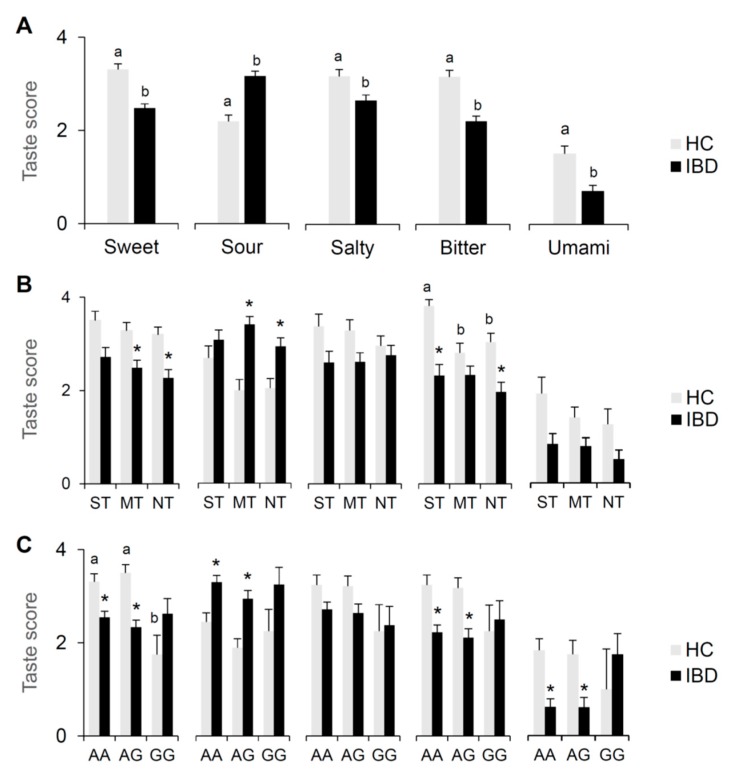
Mean values (±SE) of the taste score relative to sweet, sour, salty, bitter and umami determined in IBD patients (*n* = 97) and HC subjects (*n* = 62) (**A**). The same data are shown for each PROP taster group (**B**) and for each genotype group of the *rs2274333* (A/G) polymorphism of gustin gene (**C**). Different letters indicate a significant difference (A: *F*_1,156_ ≥ 6.1010, *p* ≤ 0.01459, one-way ANCOVA; B: *p* ≤ 0.0207; Duncan’s test adjusted by Bonferroni correction subsequent two-way ANCOVA; C: *p* ≤ 0.0297; Fisher LSD test adjusted by Bonferroni correction subsequent two-way ANOVA). * indicate a significant difference with respect to the corresponding value of HC subjects (B: *p* ≤ 0.0288; Fisher LSD test or Duncan’s test adjusted by Bonferroni correction subsequent two-way ANOVA; C: *p* ≤ 0.026; Fisher LSD adjusted by Bonferroni correction test subsequent two-way ANOVA).

**Figure 4 nutrients-12-00409-f004:**
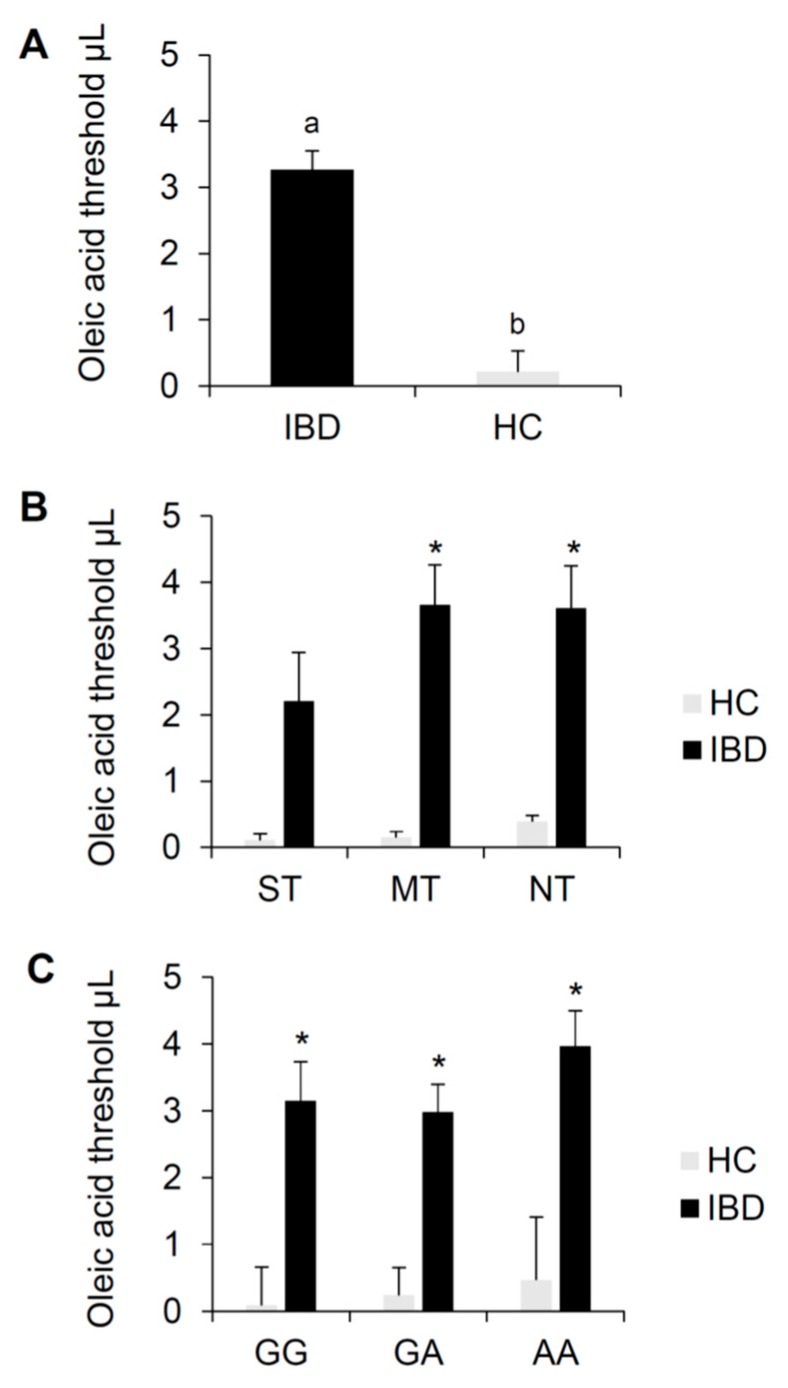
Means (±SE) values of the oleic acid threshold (µL) determined in IBD patients (*n* = 97) and HC subjects (*n* = 62) (**A**). The same data are shown for each PROP taster group (**B**) and for each genotype group of the *rs1761667* (A/G) polymorphism of *CD36* gene (**C**). Different letters indicate a significant difference (A: *F*_1,136_ = 44.779 *p* < 0.000001, one-way ANCOVA). * indicate a significant difference with respect to the corresponding value of HC subjects (B: *p* ≤ 0.00025, Duncan test subsequent adjusted by Bonferroni correction two-way ANOVA; C: *p* ≤ 0.017, Duncan test adjusted by Bonferroni correction subsequent two-way ANOVA).

**Table 1 nutrients-12-00409-t001:** Demographic and clinical features of IBD patients and HC subjects.

Variables	IBD	HC
Age (years)	51.38 ± 1.5	48.79 ± 3.06
Gender (n)		
Male	53	26
Female	44	36
CD (n)	43	-
UC (n)	54	-

IBD patients (*n* = 97) and HC subjects (*n* = 62). CD, Crohn’s disease; UC, Ulcerative Colitis.

**Table 2 nutrients-12-00409-t002:** Frequency of PROP taster groups in IBD patients and HC subjects.

Group	IBD		HC		*p*-Value
	*n*	*%*	*n*	*%*	
ST	25	25.8	14	22.6	0.741
MT	39	40.2	29	46.8	
NT	33	34	25	40.3	

*p* values derived from Fisher’s Exact Test. ST, super-tasters; MT, medium tasters; NT, non-tasters. IBD patients (*n* = 97) and HC subjects (*n* = 62).

**Table 3 nutrients-12-00409-t003:** Genotype distribution and allele frequency of polymorphisms of *Gustin* and *CD36* genes in IBD patients and HC subjects.

Polymorphisms	IBD		HC		*p*-Value
	*n*	*%*	*n*	*%*	
***Gustin gene***					
*Genotype*					
AA	53	54.6	29	46.8	0.531
AG	36	37.1	28	45.2	
GG	8	8.3	5	8	
*Allele*					
A	142	73.2	86	69.4	0.526
G	52	26.8	38	30.6	
***CD36***					
*Genotype*					
GG	18	23.7	19	30.6	0.049
AG	36	47.4	36	58.1	
AA	22	28.9	7	11.3	
*Allele*					
G	72	47.4	74	59.7	0.047
A	80	52.6	50	40.3	

*p* derived from Fisher’s method. IBD patients (*n* = 97) and HC subjects (*n* = 62).

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
