# Peer review of "Taste Changes in Patients with Inflammatory Bowel Disease: Associations with PROP Phenotypes and polymorphisms in the salivary protein, Gustin and CD36 Receptor Genes"

_nutrients, 2020, doi:10.3390/nu12020409_

Round 1
Reviewer 1 Report
The authors performed an interessant work, this is an interesting paper, well written. Authors chose an interesting topic and provided a useful contribution for clinicians and their patients. Statistical Analysis are properly performed and results are in line with the aim of the study.
Author Response
Rebuttal to comments of Reviewer
We have reworked the manuscript according to the Reviewers’ comments and suggestions.
In the revised manuscript the changes made according to Reviewer 2 are highlighted in blue, those according to Reviewer 3 in red.
Reviewer #1
The authors performed an interessant work, this is an interesting paper, well written. Authors chose an interesting topic and provided a useful contribution for clinicians and their patients. Statistical Analysis are properly performed and results are in line with the aim of the study
Reply: We really appreciate the Reviewer's nice words about the importance and potentiality of our paper.

Reviewer 2 Report
The subject is of interest with an original approach. I have only two minor comments in the introduction section :
1) Line 49 : be more accurate : which enhance intestinal barrier
2) Develop CD36 as a receptor for fat sensitivity, specially which kind of fat.
Otherwise a few mispellings
Author Response
Rebuttal to comments of Reviewer
We have reworked the manuscript according to the Reviewers’ comments and suggestions.
In the revised manuscript the changes made according to Reviewer 2 are highlighted in blue, those according to Reviewer 3 in red.
Reviewer #2
The subject is of interest with an original approach. I have only two minor comments in the introduction section:
1) Line 49 : be more accurate : which enhance intestinal barrier
Reply: We comply with the Reviewer’s request. The nutrients which enhance intestinal barrier have been added at lines 49-50.
2) Develop CD36 as a receptor for fat sensitivity, specially which kind of fat.
Reply: We comply with the Reviewer’s request. We added more details on CD36 as a receptor for fat sensitivity at lines 69-75.
Otherwise a few mispellings
Reply: we read all the manuscript and corrected the misspellings found

Reviewer 3 Report
This manuscript is an evaluation of a new approach or an attempt to correlate IBD with taste dysfunction.
Major comments:
1) I don’t see a clearly stated hypothesis. Is it along the lines of “Taste changes in IBD patients is due to specific taste dysfunction??”
2) IBD is a very broad spectrum disease and merging both UC and CD under that umbrella without even mentioning disease stages or extent is of concern especially if you are correlating with BMI. BMI in these patients tends to change over time depending on the course of the disease. In the exclusion criteria, the authors mention “major systemic diseases or any conditions or drugs interfering with taste or smell” does these also exclude IBD treatment drugs and systemic diseases related to the IBD spectra?
Minor comments:
Suggesting sections like LIMITATIONS and FUTURE DIRECTIONS separate from the main bulk of the discussion for these points are mentioned but get mixed within the points raised in the discussion sometimes.Author Response
Rebuttal to comments of Reviewer
We have reworked the manuscript according to the Reviewers’ comments and suggestions.
In the revised manuscript the changes made according to Reviewer 2 are highlighted in blue, those according to Reviewer 3 in red.
Reviewer #3
This manuscript is an evaluation of a new approach or an attempt to correlate IBD with taste dysfunction.
Major comments:
1) I don’t see a clearly stated hypothesis. Is it along the lines of “Taste changes in IBD patients is due to specific taste dysfunction??”
Reply: We recognize that the hypothesis was not clearly stated. Our work is an attempt to correlate specific dietary behaviours of IBD patients with possible taste dysfunctions. We have better stated the hypothesis at lines 95-97 and 99-100
2) IBD is a very broad spectrum disease and merging both UC and CD under that umbrella without even mentioning disease stages or extent is of concern especially if you are correlating with BMI. BMI in these patients tends to change over time depending on the course of the disease. In the exclusion criteria, the authors mention “major systemic diseases or any conditions or drugs interfering with taste or smell” does these also exclude IBD treatment drugs and systemic diseases related to the IBD spectra?
Reply: We very much thank the Reviewer for this comment which gives us the opportunity to include important information on patient’s clinical features and exclusion criteria which had been ignored. In Materials and Methods the Subjects section has been modified accordingly.
Minor comments:
Suggesting sections like LIMITATIONS and FUTURE DIRECTIONS separate from the main bulk of the discussion for these points are mentioned but get mixed within the points raised in the discussion sometimes.
Reply: We comply with the Reviewer’s request. A separate section “Limitations and Future Directions” has been added accordingly.
